# Predictive Variables for Interventional Angiography among Patients with Knee Hemarthrosis

**DOI:** 10.3390/diagnostics12040976

**Published:** 2022-04-13

**Authors:** Sang-Hun Ko, Kwang-Hwan Jung, Jae-Ryong Cha, Ki-Bong Park

**Affiliations:** Department of Orthopedic Surgery, Ulsan University Hospital, University of Ulsan College of Medicine, Ulsan 44033, Korea; jaeyun@uuh.ulsan.kr (S.-H.K.); jkh2007@uuh.ulsan.kr (K.-H.J.); jrcha@uuh.ulsan.kr (J.-R.C.)

**Keywords:** hemarthrosis, synovial fluid, analysis, angiography, embolization, therapeutic

## Abstract

Studies regarding the variables that could predict the success of conservative treatment for knee hemarthrosis are lacking. This retrospective study evaluated the laboratory variables of patients who had unsatisfactory results from conservative treatment for knee hemarthrosis. Twenty-nine patients conservatively treated for knee hemarthrosis were included and divided into two groups: group A comprised 14 patients who underwent interventional angiography and selective embolization due to failed conservative treatment, and group B comprised 15 patients with successful results after conservative treatment. The results of the serological and synovial fluid tests were evaluated. The mean number of synovial red blood cells (RBCs) was 1,905,857 cells/µL and 7730 cells/µL in groups A and B, respectively (*p* = 0.01), while the mean number of RBCs per high-power field (HPF) was 68.9 and 3.2, respectively (*p* < 0.01). Patients who underwent interventional angiography and selective embolization after failed conservative treatment for knee hemarthrosis had higher synovial RBC counts and RBC counts per HPF than those with successful outcomes after conservative treatment. It is necessary to carefully interpret the results of the synovial fluid analysis in patients with knee hemarthrosis; if the synovial fluid analysis shows a synovial RBC count greater than 81,500 and RBC count per HPF greater than 16.3, we recommend immediate interventional angiography rather than continuing conservative treatment.

## 1. Introduction

Several disorders can present with hemarthrosis of the knee joint [1,2,3], which can be divided into traumatic, postoperative, and non-traumatic categories [4]. Traumatic injury is the most common cause of knee hemarthrosis [5,6,7]. Postoperative hemarthrosis is frequently associated with total knee arthroplasty (TKA) and has also been described as an uncommon complication following arthroscopy [2,8,9,10]. Non-traumatic hemarthrosis can be broadly divided into congenital and acquired types [11]. After exclusion of coagulopathy, other causes, such as arthritis, collagen vascular disorders, hemochromatosis, myeloproliferative diseases, and vascular lesions, need to be considered [12,13,14,15,16].

The diagnosis of hemarthrosis may be based on a suggestive history, physical examination, or imaging studies; however, definitive diagnosis usually requires arthrocentesis [17]. Arthrocentesis can be used to identify and diagnose hemarthrosis by ruling out simple effusion, lipo-hemarthrosis, and infection or inflammatory arthritis. Hemarthrosis is diagnosed if arthrocentesis shows gross bloody features without fat globules. Among the various imaging studies, computed tomography (CT) angiography or interventional angiography has been used as a diagnostic tool to identify the cause of hemarthrosis [2,17]. However, in patients with hemarthrosis after TKA, CT angiography may not be helpful because of the interference effect of the artifact produced by the prosthesis and the small size of the blood vessels where the actual bleeding site is located [17].

Treatment methods for knee hemarthrosis include conservative therapy by resting, immobilizing the joint in a cast or brace, applying an ice pack, and performing arthrocentesis repeatedly [18]. The proportion of patients with hemarthrosis recurrence despite conservative treatment is small [11,19]. After failed conservative treatment for knee hemarthrosis, interventional angiography is used to aid in the diagnosis, and concomitant selective embolization is performed to control bleeding [11,17,19,20,21,22].

We aimed to determine the variables that could predict the necessity for interventional angiography for knee hemarthrosis, as it would be helpful in the early detection and management of patients who tend to fail conservative treatment. Therefore, we reviewed our experience with patients who had knee hemarthrosis and underwent it with or without interventional angiography. We hypothesized that among the laboratory variables of patients who did not obtain satisfactory results from conservative treatment for knee hemarthrosis and underwent interventional angiography, there could be predictive factors regarding the indications for interventional angiography. Therefore, this study evaluated the laboratory variables of patients who exhibited unsatisfactory results from conservative treatment for knee hemarthrosis.

## 2. Materials and Methods

### 2.1. Study Population

In this retrospective study, we included 29 patients (29 knees) who were diagnosed with knee hemarthrosis based on medical history, clinical symptoms, physical examination, and arthrocentesis at a single tertiary referral hospital between 2001 and 2020. The mean age was 67.9 years (range 51–85 years) and sex ratio (male:female) was 14:15. We excluded patients with systemic coagulopathy (such as hemophilia), those who were being administered anticoagulant or antiplatelet medications, and those with intra-articular injuries, such as cruciate ligament rupture, meniscus tear, and intra-articular fractures.

All included patients were started on conservative treatment, including immobilization of the joint, application of ice packs, compression, and discontinuation of anticoagulants or antiplatelet drugs. We performed arthrocentesis and analyzed the synovial fluid obtained through the first arthrocentesis. We defined recurrent hemarthrosis as a case with hemarthrosis in secondary arthrocentesis performed due to persistent effusion after 4–6 weeks of conservative treatment at our institution. Depending on whether the patient had recurrent hemarthrosis after conservative treatment, CT angiography and interventional angiography were performed. Informed consent for our treatment protocol was obtained from all the patients. Our institutional review board approved this study (UUH-2021-12-036).

We classified patients into two groups: group A, consisting of 14 patients who did not obtain satisfactory results from conservative treatment and underwent interventional angiography and selective embolization, and group B consisting of 15 patients who obtained satisfactory results from conservative treatment (Figure 1). We compared the demographic information, serologic tests, aspirated synovial fluid analyses, and clinical outcomes of both groups and investigated interventional angiographic findings and procedures related to embolization in group A.

### 2.2. Evaluation

Demographic information included age, sex, and surgical history of the affected knee. Serologic tests included hemoglobin, hematocrit, prothrombin time (PT), activated partial thromboplastin time, international normalized ratio, erythrocyte sedimentation rate, and C-reactive protein. We defined anemia as hemoglobin levels < 12.0 g/dL in women and <13.0 g/dL in men. Analysis of aspirated synovial fluid included the synovial red blood cell (RBC) count, synovial white blood cell (WBC) count, percentage of polymorphonuclear (PMN) cells, and number of RBCs per high-power field (HPF) under smear. Interventional angiography and selective embolization were performed by the radiologist. We evaluated the findings of the interventional angiography (location of blood vessels and abnormalities such as bleeding, hypervascularity, and pseudoaneurysm) and materials used for embolization. After discharge, no specific follow-up protocol was required. However, clinical examinations and/or imaging studies were performed if patients reported symptoms of recurrent hemarthrosis. Clinical success was defined as no recurrence of hemarthrosis within the follow-up period.

### 2.3. Statistical Analysis

All measurements were expressed as mean (range), and independent t-tests were performed using SPSS for Windows version 11.5 (SPSS Inc., Chicago, IL, USA). Receiver operating characteristic curve analysis using the Youden index was conducted to determine an optimal cut-off of total synovial RBC count and RBC counts per HPF for predicting the need for interventional angiography. *p* < 0.05 was considered statistically significant.

## 3. Results

### 3.1. Description of the Study Population and Their Surgical History

The characteristics of the study population are summarized in Table 1. There was no difference between the two groups in age, total period of conservative treatment, and sex ratio. Among 29 patients, 17 patients (58.6%) developed hemarthrosis in the operated knee. Patients with postoperative knee included patients who underwent TKA and revised TKA, and none of the patients had a recent history of trauma. Consequently, all patients who were included in this study had no recent history of trauma. We further analyzed the differences between patients with native knees and patients with postoperative knees within each group and confirmed that there was no significant difference between the native and postoperative knee groups. The mean number of arthrocentesis procedures was 5.1 (range 4–6) in group A and 3.3 (range 3–4) in group B (*p* < 0.01). The mean number of arthrocentesis procedures was significantly higher in group A than in group B.

### 3.2. Comparison of Serologic Test Results and Synovial Fluid Analyses

Table 2 compares the serologic test results and synovial fluid analyses between the two groups. There were no significant differences in the levels of hemoglobin (10.3 g/dL and 10.7 g/dL) and hematocrit (31.0% and 33.1%) between the two groups (*p* = 0.66, 0.45). The percentage of patients with anemia was 75.6% (11/14) in group A and 73.3% (11/15) in group B (*p* = 0.75). The mean number of RBCs in the synovial fluid was significantly higher in group A (1,905,857 cells/µL) than in group B (7730 cells/µL) (*p* < 0.01). There were no significant differences in the mean number of WBCs and PMN cells in the synovial fluid between the two groups.

On the synovial fluid smear, the mean number of RBCs per HPF in group A was 68.9 (range 25–100), and it was significantly higher than the 3.2 (range 0.5–25) found in group B (*p* < 0.01) (Figure 2).

### 3.3. Angiographic Findings and Follow-Up

Table 3 shows the interventional angiographic findings in group A. The lateral inferior, lateral superior, and medial descending genicular arteries were commonly identified as feeding vessels on interventional angiography. The dominant findings of the interventional angiography were hypervascularization and contrast blush. Selective embolization was performed in all patients, which contributed to successful hemostasis. Selective embolization was performed using gelfoam (n = 10) or particle materials (n = 4).

Patients visited the outpatient clinic for regular follow-up or were contacted by telephone in December 2021 to retrieve information on recovery or recurrence. There were no recurrences of hemarthrosis in any group. The mean duration of follow-up after CT angiography was 10.2 years (range 1.1–19.8 years).

## 4. Discussion

The major finding of this study was that the synovial RBC count and number of RBCs per HPF under the smear correlated with the failure of conservative treatment in knee hemarthrosis and thus may be used to predict the necessity of interventional angiography for further evaluation and treatment. We interpreted total synovial RBC count greater than 81,500 and RBC counts per HPF greater than 16.3 in the synovial fluid analysis as suggestive of potential conservative treatment failure.

It is well known that the synovial WBC count and percentage of synovial PMN cells are useful for distinguishing between non-inflammatory, inflammatory, and infectious arthritis [23,24]. However, the significant findings to note in aspirated synovial fluid from patients diagnosed with knee hemarthrosis remain uncertain. In a study analyzing preoperative synovial fluid in patients undergoing revision TKA for flexion instability, Hernandez et al. included the total WBC count, percentage of neutrophils, and the appearance of the synovial fluid as variables for comparison [25]. They reported that there was significantly more bloody synovial fluid in the flexion instability group than in the control group. Although they considered bloody serosanguineous aspirations to be consistent with hemarthrosis, they did not describe the synovial RBC count in the analysis of synovial fluid. In this study, synovial fluid analysis included all variables that were reported previously and showed that the patients who underwent interventional angiography after failed conservative treatment for knee arthrosis had a higher synovial RBC count and number of RBCs per HPF under synovial fluid smear following arthrocentesis.

The diagnosis of hemarthrosis is predominantly based on the combination of clinical symptoms and arthrocentesis [18,26]. Waldenberger et al. investigated 35 patients with knee hemarthrosis and confirmed knee hemarthrosis by aspiration of bloody synovial fluid in 12 patients and by magnetic resonance imaging in the remaining patients [20]. Bagla et al. investigated five patients with hemarthrosis after TKA and only described hemorrhagic effusion in repeated arthrocentesis [21]. Park et al. investigated seven patients with hemarthrosis after TKA and diagnosed hemarthrosis using medical records, patients’ symptoms, and arthrocentesis [17]. However, they analyzed aspirated synovial fluid to rule out infections or inflammatory arthritis. In this study, we evaluated aspirated synovial fluid analysis and described the detailed results including synovial RBC count, WBC count, percentage of synovial PMN cells, and number of RBCs per HPF under the smear. We also showed that there were significant differences in the synovial RBC count and number of synovial RBCs per HPF under the smear between the groups.

Few studies have investigated the hemodynamic status of patients with knee hemarthrosis. A retrospective study of 35 patients with knee hemarthrosis reported that none of the patients had declined hemoglobin levels but did not provide detailed information on the serologic tests [20]. In this study, we evaluated the serologic tests and described the values. There were no differences in the levels of hemoglobin and hematocrit or the percentage of patients with anemia in both groups.

Several studies have investigated interventional angiographic findings in patients who underwent conservative treatment for knee hemarthrosis and interventional angiography. A retrospective study of 25 patients undergoing angiography for knee hemarthrosis reported that the identified vascular abnormalities were periarticular synovial hypervascularity (92%) and pseudoaneurysm (8%) [27]. A case series of seven patients with hemarthrosis after TKA reported that the main findings of interventional angiography were pathologic vascular blush (85.7%) and pulsatile bleeding (14.3%) [17]. In this study, the main findings in 14 patients who underwent interventional angiography were hypervascularity and contrast blush. Interventional angiography was considered a useful investigation for diagnosing the cause of bleeding in both early and late cases of knee hemarthrosis. In the early presenting cases, pseudoaneurysm and arteriovenous fistulas were usually identified, and in later cases, vascular blush appeared to indicate synovial hypertrophy [2]. Since previous studies [17,27], including this study, have included patients whose knee hemarthrosis has not resolved despite conservative treatment, we inferred that the main findings of interventional angiography in these patients were vascular blush or synovial hypertrophy, which was found in late cases of knee hemarthrosis.

The advantages of interventional angiography are that various treatment options, including stenting and embolization, can be performed at the time of the diagnostic procedure [2]. A retrospective study of 25 patients undergoing angiography for knee hemarthrosis reported that the most commonly embolized vessels were the medial superior genicular artery and lateral superior genicular artery [27]. A case series of seven patients with hemarthrosis after TKA reported that six cases demonstrated bleeding from the inferior lateral genicular artery and one from the middle genicular artery [17]. In this study, the common feeding vessels were the lateral inferior genicular artery and lateral superior genicular artery.

In this study, the symptoms of patients in group B improved with only conservative treatment; however, those of patients in group A did not improve despite undergoing conservative treatment for a similar period and only improved after interventional angiography and selective embolization. We believe that the difference in the severity of knee hemarthrosis between the two groups may have caused the difference in the outcomes of conservative treatment. In this study, the number of arthrocentesis procedures was significantly higher in group A than in group B; hence, we estimated that bleeding of unknown cause was continuously occurring in the knee joint of patients in group A. We assumed that the possibility of continuous bleeding was valid to some extent as findings, such as hypervascularity, contrast blush, and pseudoaneurysm, were confirmed in the interventional angiography performed in group A. Moreover, since patients in group A showed improvement in their symptoms after selective embolization, we believe that they had knee hemarthrosis that was more difficult to treat than those of patients in group B, whose symptoms improved with only conservative treatment. All patients in this study achieved clinical success within the follow-up period. In previous studies, rates of technical success with embolization were reported to be 99–100% with clinical success rates of 80–93% [20,28,29,30,31].

The limitations of this study include the relatively small number of patients, discrepancies in severity, failing to present some data, and long-time span of observation. Since all patients were diagnosed and managed at one hospital, the number of patients enrolled in this study was small. Nevertheless, this study included patients who presented with knee hemarthrosis and underwent conservative treatment at one hospital over a long study period (21 years). It could be assumed that the patients in group B were initially less symptomatic because of the smaller number of arthrocentesis procedures compared to that of group A. Therefore, care should be taken when interpreting the results of this study, as the less severe symptoms may have been potential confounding factors that could have affected the results of conservative treatment in knee hemarthrosis. Third, the amount of blood aspirated during each arthrocentesis could not be presented in this study due to insufficient data. Finally, because this study had a long study period, there may be disadvantages such as change in therapeutic strategy and novel anticoagulation therapy.

## 5. Conclusions

Patients who underwent interventional angiography and selective embolization after failed conservative treatment for knee hemarthrosis had higher total synovial RBC counts and RBC counts per HPF than those who had successful outcomes after conservative treatment. It is necessary to carefully interpret the results of the synovial fluid analysis in patients with knee hemarthrosis, and if the synovial fluid analysis shows a total synovial RBC count greater than 81,500 and RBC counts per HPF greater than 16.3, we recommend immediate interventional angiography rather than continuing conservative treatment.

## Figures and Tables

**Figure 1 diagnostics-12-00976-f001:**
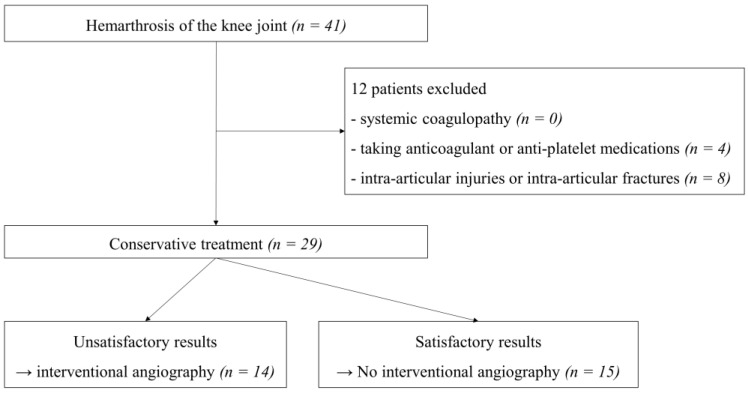
Flow chart of study population selection.

**Figure 2 diagnostics-12-00976-f002:**
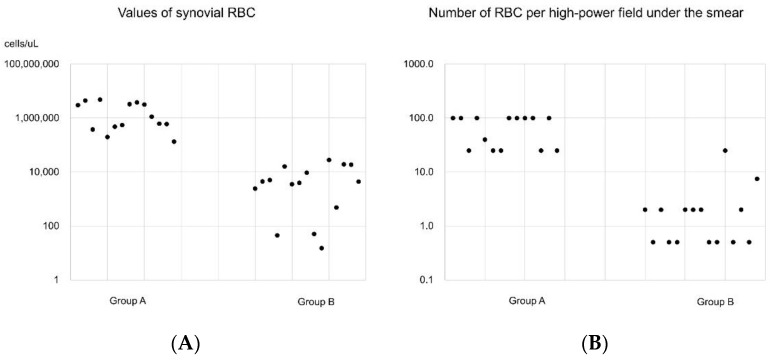
(**A**) Values of synovial red blood cells (RBCs) and (**B**) number of RBCs per high-power field under the smear of the synovial fluid smear. Group A: interventional angiography and selective embolization after failed conservative treatment. Group B: successful results after conservative treatment.

**Table 1 diagnostics-12-00976-t001:** Clinical characteristics of 16 patients who were diagnosed with knee hemarthrosis and underwent computed tomography angiography.

Variables	Group A (n = 14)	Group B (n = 15)	*p*-Value
	Mean (range)		
Age (years)	67.2 (52–85)	68.5 (51–83)	0.73
Duration of conservative treatment (months)	4.3 (2.4–7.5)	3.9 (2.8–5.1)	0.21
Number of arthrocentesis procedures	5.1 (4–6)	3.3 (3–4)	<0.01
Follow-up period (months)	22.3 (12.6–38.2)	21.2 (12.3–50.6)	0.75
	Number		
Sex (female:male)	8:6	6:9	0.37
Surgical history			
None (native knee)	5	7	
TKA	5	7	
Revision TKA	4	1	

Group A: interventional angiography and selective embolization after failed conservative treatment; Group B: successful results after conservative treatment; TKA, total knee arthroplasty.

**Table 2 diagnostics-12-00976-t002:** Serologic test results and synovial fluid analyses between two groups.

Variables	Group A (n = 14)	Group B (n = 15)	*p*-Value
Serum			
Hb (g/dL)	10.3 (5.3–14.9)	10.7 (6.3–13.9)	0.66
Hct (%)	31.0 (16.1–44.3)	33.1 (18.7–43.1)	0.45
PT (s)	12.5 (10.7–14.3)	11.9 (10.3–13.4)	0.14
aPTT (s)	30.9 (24.3–38.8)	31.7 (24.9–36.7)	0.57
INR	1.1 (0.9–1.5)	1.0 (0.9–1.2)	0.20
ESR (mm/h)	12.5 (2–20)	14.7 (2–24)	0.37
CRP (mg/dL)	0.5 (0.1–1.0)	0.6 (0.1–1.1)	0.19
Synovial Fluid			
RBC (cells/µL)	1,905,857 (200,000–4,890,000)	7730 (15–28,000)	<0.01
WBC (cells/µL)	1386.5 (4–4500)	2230.0 (140–4160)	0.15
Neutrophil (%)	39.6 (13–52)	32.6 (10–69)	0.25
RBC (number/HPF)	68.9 (25–100)	3.2 (0.5–25)	<0.01

Group A: interventional angiography and selective embolization after failed conservative treatment; Group B: successful results after conservative treatment; Hb, hemoglobin; Hct, hematocrit; PT, prothrombin time; aPTT, activated partial thromboplastin time; INR, international normalized ratio; ESR, erythrocyte sedimentation rate; CRP, C-reactive protein; HPF, high-power field.

**Table 3 diagnostics-12-00976-t003:** Detailed information on patients who underwent interventional angiography and selective embolization.

Age (Years)/Sex	Surgical History	Feeding Artery	Findings
75/Male	TKA	LSGA, DGA	Hypervascularization
59/Male	TKA	LIGA	Hypervascularization, contrast blush
67/Male	TKA	LIGA, anterior tibial recurrent artery,	Contrast blush
53/Male	None	Medial DGA	Hypervascularization
71/Male	None	LIGA, LSGA, Medial DGA	Hypervascularization, contrast blush
52/Male	None	LIGA, LSGA	Hypervascularization
72/Female	Revision TKA	LIGA	Hypervascularization, contrast blush
78/Female	Revision TKA	LSGA, MSGA, muscular artery	Multiple pseudoaneurysm, contrast blush
85/Female	Revision TKA	Medial DGA, anterior tibial recurrent artery	Hypervascularization, contrast blush
74/Female	Revision TKA	Multiple genicular artery	Hypervascularization, contrast blush
75/Male	TKA	Medial DGA, LSGA	Hypervascularization
63/Female	None	Medial DGA	Hypervascularization
60/Male	None	LIGA, LSGA	Hypervascularization
57/Female	TKA	Medial DGA, LIGA	Hypervascularization, contrast blush

TKA, total knee arthroplasty; LSGA, lateral superior genicular artery; DGA, descending genicular artery; LIGA, lateral inferior genicular artery; MSGA, medial superior genicular artery.

## Data Availability

Not applicable.

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
