# Peer review of "Predictive Variables for Interventional Angiography among Patients with Knee Hemarthrosis"

_diagnostics, 2022, doi:10.3390/diagnostics12040976_

Round 1

Reviewer 1 Report

Thank you very much for letting me review your manuscript. I appreciate the effort you have made performing this paper and submitting it to Diagnostics. The review process aims to improve your manuscript, even when it might not be suitable for publication here. Please meticulously consider the comments and feedback given in the following. Do not take it personally, as the only aim is to improve your scientific manuscript.

The current manuscript aims to determine the variables that could predict the necessity of interventional angiography for knee hemarthrosis. Thereby the authors investigated patients with recurrent hemarthrosis in their clinics and created two groups: group A: failed conservative treatment with following interventional angiography; B: successful conservative treatment. The following parameters were compared: demographic information, serologic tests, aspirated synovial fluid analyses, and clinical outcomes of both groups. Patients with failed conservative treatment had higher synovial RBC and RBC counts per HPF under smear than those of patients with successful results after conservative treatment. In general the study is of clinical importance, but the study has some major and minor issues that have to be addressed. The selection process has to be clarified in more detail. The question if there was a mixing up of traumatic/atraumatic and postoperative knee hemarthrosis has to be answered and if so addressed. It has to be clarified which of the arthrocentesis was used for comparison since all patients received multiple arthrocentesis. Finally, the clinical value of the present study remains unclear since no threshold was presented for the data. The clinical value and possible clinical consequences should more precisely be addressed in the discussion.

Abstract

Line 10: What was the study design? Retrospective or prospective? Please add.

Line 21 -23: Please add a conclusion that can be done because of your particular results and do not only write that careful interpretation is necessary.

What can we learn from your study?

Keywords

ok

Title

ok

Introduction

Well written Introduction.

Line 55: I don’t understand - Did all patients receive CT-Angiography? Did not all patients that receive CT-Angiography receive embolisation? Please be more precise here.

Material and methods

What was the study design? Retrospective or prospective? Please add.

Line 65-66: The time span of your observation = 19 years which is quite long. How do you think that the time span might have biased your observation? (e.g. change of therapeutic strategy, novel anticoaglation therapy). Please discuss and outline how you possibly have anticipated this.

Linen 72-74: How do you define recurrent hemarthrosis after conservative treatment? Was a second hemorthros after arthrocentesis  + conservative treatment already recurrent for you?

Line 78: Please outline the inclusion and exclusion criteria in detail: Did you include traumatic, atraumatic, postoperative patients. What about age range?

It is very important if you mixed traumatic/atraumatic patients with post-operative patients if those are completely different patient cohort. Do not mix these cohorts and investigate both groups separately to see if there are differences. Please comment here.

Line 90: In table 1 we can see that the range of arthrocentesis was 2-4. In that case what arthrocentesis did you use for analysis? Second one? Or the last one? Did you have a systematic approach? If not please discuss this in your limitations section.

Line 99: How long did you follow-up the patients? Did you follow-up all the patients from both groups?

Results

Line 105-115: You do not have to give the demographic information in written form and in the tables. Please only give the most important information here and refer to the table.

Discussion

Is there a threshold of RBCs that you could recommend that to indicate failed conservative treatment and indication for embolisation? Please discuss.

Line 166: What were their findings of RBC? Please discuss the findings in the context with your present study.

Line 230: Did you measure the amount of blood that you aspirated from each arthrocentesis? Otherwise how can you define severity? Please comment.

Line 232: What was the follow-up period?

Line 240: Please see above. What might the disadvantages be of that long study period?

Conclusion

Lin 256: Can you provide a conclusion? Can you suggest a threshold? What can we learn from your study? What is the clinical value? 

Reviewer 2 Report

Interesting paper that points possible solutions on the treatment of hemarthrosis resistant to conservative treatment.

It is an investigating looking for solutions, giving some clues, but need to be validated by other works

Author Response

Thank you for reviewing our manuscript.

We will revise the manuscript according to the comments of another reviewer.

Round 2

Reviewer 1 Report

Thank you very much for your efforts in revising the mansucript.

You provided the following conclusion:

“if the synovial fluid analysis of patients with knee hemarthrosis shows greater than 81,500 RBCs in the
synovial fluid and RBC count per HPF greater than 16.3, we recommend immediate interventional angiography
rather than continuing conservative treatment.” (Line 2224)
"

Question: On what basis did you set this threshold? An appropriate justification is very important.

Author Response

Question: On what basis did you set this threshold? An appropriate justification is very important. (Line 22–24)

Answer: We described that receiver operating characteristic curve analysis was performed using Youden index to find out the optimal cut-off. (Line 108–111)